# Peering through the PSMA PET Lens: The Role of the European Association of Urology Biochemical Recurrence Risk Groups after Radical Prostatectomy

**DOI:** 10.3390/cancers15112926

**Published:** 2023-05-26

**Authors:** Charles Leplat, Teddy Jabbour, Romain Diamand, Arthur Baudewyns, Henri Alexandre Bourgeno, Qaid Ahmed Shagera, Patrick Flamen, Thierry Roumeguere, Alexandre Peltier, Carlos Artigas

**Affiliations:** 1Department of Urology, Institut Jules Bordet, Erasme Hospital, Hôpital Universitaire de Bruxelles, Université Libre de Bruxelles, 1070 Brussels, Belgium; charles.leplat@gmail.com (C.L.); teddy.jabbour@hubruxelles.be (T.J.); romain.diamand@hubruxelles.be (R.D.); arthur.baudewyns@hubruxelles.be (A.B.); henri.bourgeno@hubruxelles.be (H.A.B.); thierry.roumeguere@hubruxelles.be (T.R.); alexandre.peltier@hubruxelles.be (A.P.); 2Nuclear Medicine Department, Institut Jules Bordet, Erasme Hospital, Hôpital Universitaire de Bruxelles, Université Libre de Bruxelles, 1070 Brussels, Belgium; qaid.shagera@ulb.be (Q.A.S.); patrick.flamen@hubruxelles.be (P.F.)

**Keywords:** PSMA, biochemical recurrence, prostate cancer, oligometastasis, EAU risk group

## Abstract

**Simple Summary:**

After radical prostatectomy (RP), about a third of patients present with BCR. The European Association of Urology (EAU) biochemical recurrence (BCR) risk grouping relies on data from historical cohorts that used conventional imaging techniques. Given the importance of differentiating the recurrence patterns and their impact on treatment decisions, the indication for PSMA PET/CT has been established in BCR prostate cancer. The main aim of this study was to study the positivity rate of ^68^Ga-PSMA-11 PET/CT across BCR low and high-risk groups and to establish positivity predictors. We found a higher rate of positivity within the BCR high-risk group, whereas more local and oligometastatic recurrences were objectified in BCR low-risk group. This raises the discussion for including PSMA PET/CT within the risk stratification after carefully evaluating positivity predictive factors. Future studies are needed to confirm the above findings for better patient selection and management.

**Abstract:**

(1) Background: The European Association of Urology (EAU) biochemical recurrence (BCR) risk grouping relies on data from historical cohorts that used conventional imaging techniques. In the era of PSMA PET/CT, we compared the patterns of positivity in the two risk groups and provided insight into positivity predictive factors. (2) Methods: Data from 1185 patients who underwent ^68^Ga-PSMA-11PET/CT for BCR was analyzed, out of which 435 patients treated initially treated by radical prostatectomy were included in the final analysis. (3) Results: A significantly higher rate of positivity in the BCR high-risk group was observed (59% vs. 36%, *p* < 0.001). BCR low-risk group demonstrated more local (26% vs. 6%, *p* < 0.001) and oligometastatic (100% vs. 81%, *p* < 0.001) recurrences. The BCR risk group and PSA level at the time of PSMA PET/CT were independent predictive factors of positivity. (4) Conclusions: This study confirms that the EAU BCR risk groups have different rates of PSMA PET/CT positivity. Even with a lower rate in the BCR low-risk group, oligometastatic disease was 100% in those with distant metastases. Given the presence of discordant positivity and risk classification, integrating PSMA PET/CT positivity predictors into risk calculators for BCR might improve patient classification for subsequent treatment options. Future prospective studies are still needed to validate the above findings and assumptions.

## 1. Introduction

Prostate cancer (PCa) is the second most common male cancer worldwide. This represents approximately 1.3 million new cases along with approximately 350,000 deaths yearly in 2020 [1]. Given the substantial burden of this disease worldwide, significant innovations have been put forth to aid in the diagnosis and treatment of upfront disease and recurrent disease after initial management. 

After initial treatment with radical prostatectomy (RP) for localized disease, more than a third of patients experience biochemical recurrence (BCR) [2], defined as two successive PSA values >0.2 ng/mL [3]. This represents a significant subset of the patient population that might benefit from enhanced early risk stratification to guide future management options. 

Accurate diagnosis of disease recurrence is essential to distinguish between local recurrence, oligometastatic disease, and extensive metastatic disease, given that the management and prognosis are radically different [4,5,6,7]. Metastatic patients will require systemic treatment with androgen deprivation therapy (ADT) in association with androgen receptor-targeted agents or chemotherapy, whereas oligometastatic patients may benefit from metastasis-directed therapies (MDT). Although the validation of such therapies in this setting is still under evaluation, MDT has recently gained momentum as a means of controlling disease progression while delaying the need for systemic treatment, thereby avoiding its toxicity [8,9].

The early detection of oligometastatic patients through the use of imaging techniques is crucial to initiate targeted treatments promptly. Prostate-specific membrane antigen (PSMA) is a transmembrane glycoprotein overexpressed by prostate cancer cell membranes [10]. PCa cells exhibit overexpression of this particular protein in over 90% of cases, yet its precise role remains unclear. Nonetheless, its involvement in enhancing tumor growth, migration, and invasion suggests its utility as a diagnostic and therapeutic modality. Radiotracers have been developed based on peptides that actively bind to the extracellular domain of PSMA, coupled with a positron-emitting isotope such as gallium-68 or fluorine-18 [11], allowing whole-body positron emission tomography (PET)/computed tomography (CT) imaging. Since the discovery of this new technology, indications for this examination have multiplied. 

Indeed, PSMA PET/CT has better accuracy compared to MRI for preoperative staging of intermediate- and high-risk prostate cancer as well as at BCR [12]. Moreover, PSMA PET/CT is more sensitive than the conventional metastatic workup by CT scan and bone scintigraphy [13] and more accurate than ^18^F-Choline-PET/CT [14] and ^18^F-Fluciclovine-PET/CT [15]. Nevertheless, randomized studies are needed to evaluate the impact of this examination on the management and prognosis of patients. 

The European Association of Urology (EAU) recently defined predictive risk groups for progression during BCR (Table 1) based on disease characteristics and PSA kinetics [16,17]. This risk grouping is based on data from historical groups that used traditional imaging techniques, including CT scans and/or whole-body bone scans, to evaluate the spread of cancer. The aim of this study is to compare the rate and patterns of positivity of PSMA PET/CT in the two risk groups. We then provide insight into predictive factors for positivity.

## 2. Materials and Methods

### 2.1. Population

The study was performed following the Declaration of Helsinki after obtaining the institutional review board’s approval. Data from a prospectively maintained database of 1185 patients who underwent ^68^Ga-PSMA-11 PET/CT scans to investigate cancer recurrence between 2015 and 2021 were collected. We selected patients that presented with BCR after RP according to the EAU guidelines [3]. The status of receiving adjuvant and salvage radiotherapy (sRT) was recorded. Patients were not on ADT at the time of the PSMA PET/CT. Patients presenting with biochemical persistence and castration-resistant prostate cancer (CRPC) were excluded. The final cohort consisted of 435 patients eligible for analysis. 

### 2.2. Radiotracer Preparation

^68^Ga was obtained by elution of a ^68^Ge/^68^Ga radionuclide generator. It was used for radiolabeling after 5 min of incubation at room temperature using a sterile cold kit containing 25 µg of a lyophilized PSMA-11 precursor following the manufacturer’s instructions. Quality control was performed using fine chromatography and showed a radiochemical purity of over 99%. Sterility and pyrogen testing was performed per European Pharmacopoeia methods.

### 2.3. Imaging Procedure

Images were obtained using a Discovery 690 TOF PET system (General Electric, Milwaukee, WI, USA) 60 min after an injection of 2 Mbq/kg of ^68^Ga-PSMA-11. Patients emptied their bladder immediately before the scan and were not required to fast or follow a specific diet. Patients were scanned from the mid-thigh to the top of the skull. All PET scans were acquired in 3D mode with an acquisition time of 2 min/step with a 23.4% overlap. A low-dose CT (120 kV) was performed without contrast injection.

### 2.4. Image Analysis

All images from PSMA PET/CT were analyzed by two experienced nuclear medicine physicians. The interpretation was performed according to the standards of the European Association of Nuclear Medicine (EANM), and any focal uptake of ^68^Ga-PSMA-11 not associated with physiological uptake was considered suspicious for malignancy [18]. The location and number of lesions were specified. Positive lesions were validated by diagnostic imaging, histology, and/or clinical follow-up. A multidisciplinary oncology concertation, consisting of at least one urologist, a nuclear medicine physician, a radiation oncologist, a radiologist, and a medical oncologist, decided by consensus appropriate patient management.

### 2.5. Variables and Outcomes

PSA, as well as PSA kinetics (i.e., PSA doubling time (PSAdt), PSA velocity (PSAvel) using the MSKCC nomogram), were measured at the time of the PSMA PET/CT. Other variables, including age, TNM stage, International Society of Urological Pathology (ISUP) grade group, margins status after RP, adjuvant treatment, and sRT, were collected. We defined oligometastasis according to the ESTRO-ASTRO consensus, which accepts up to 5 positive lesions [19].

The primary outcome of this study was the positivity rate of PSMA PET/CT between the EAU BCR risk groups. The pattern of positivity was also analyzed. The secondary outcome was the evaluation of positivity predictive factors across the risk groups.

### 2.6. Statistical Analysis

Analysis of demographic data was performed using continuous variables and calculated medians and interquartile ranges (IQR). Categorical variables were expressed as proportions. The Mann-Whitney U-test was used to compare medians, while chi-square tests were applied to compare proportions. We searched for predictive factors of positivity by performing univariable and multivariable logistic regressions and defined cut-offs using ROC curves. A sensitivity analysis was conducted to evaluate whether the association between the variables and the outcomes varied according to the oligometastatic definition. A sub-analysis was performed for positivity predictors for BCR low-risk patients. Any *p*-value < 0.05 was considered statistically significant. Statistical analysis was performed using SPSS statistics v28 (IBM Corp, Armonk, NY, USA).

## 3. Results

### 3.1. Patient Characteristics

The clinical and pathological characteristics of the patients are found in Table 2. The median serum PSA at the time of PSMA PET/CT was 1.14 ng/mL (IQR 0.5–3). The median PSAdt and PSAvel were 6.7 (IQR 3.9–13) months and 0.9 (IQR 0.3–2.8) ng/mL/year, respectively. This was the first BCR (i.e., only RP or RP followed by adjuvant RT) in 235 (54%) of patients, while 200 (46%) patients had already received sRT. Overall, 97 (22%) patients were classified in the BCR low-risk group, and 338 (78%) patients in the BCR high-risk group.

### 3.2. Imaging Findings

PSMA PET/CT identified at least one PSMA-overexpressing lesion that raised the suspicion for clinical recurrence of PCa in 54% (236/435) of the patients who were part of the study. The majority of the patients (51%) had pelvic disease defined as either local (i.e., surgical bed) or regional nodal metastasis (N1). Extra-pelvic nodal involvement (M1a) was observed in 18% (43/236) of patients. Bone involvement was noted in 25% (59/236) of patients, and visceral metastasis was found in 5.9% (14/236) of patients, as indicated in Table 3 and Figure 1 and Figure 2. Of these visceral metastases, the majority, 76% (11/14), were pulmonary. Among patients with distant metastatic disease, 100% were oligometastatic in the BCR low-risk group (Figure 3) compared to 81% in the BCR high-risk group (*p* < 0.001). Moreover, a significantly higher percentage of local recurrence was found in the BCR low-risk group (26% vs. 6%, *p* < 0.001).

Multivariable logistic regression analysis showed that the BCR risk group (OR 2.2; 95% CI 1.3–3.8; *p* = 0.004) and PSA level at the time of PSMA PET/CT (OR 2.7; 95% CI 1.7–4.4; *p* < 0.001) were independent predictive factors of positivity. In the BCR low-risk group, PSA at the time of PSMA PET/CT and PSAvel were significantly associated with a higher probability of positivity (Table 4). Multivariate analysis for BCR low-risk could not be performed due to a lack of statistically significant independent covariables.

## 4. Discussion

In the present study, we demonstrated the clinical value of the EAU BCR risk groups by showing a higher percentage of ^68^Ga-PSMA-11 PET/CT positivity in BCR high-risk patients. However, our results also demonstrate that BCR low-risk patients may also benefit from this imaging modality given the high percentage of oligometastatic disease detection. 

Detecting oligometastatic PCa recurrences enables metastasis-directed therapy, the prognostic value of which appears to be clinically promising. Published pooled data from two trials (STOMP and ORIOLE) demonstrates that metastasis-directed therapy improves progression-free survival from 5.9 months (95% CI: 3.2–7.1) to 11.9 months (95% CI: 8.0–18.3; HR: 0.44, *p* < 0.001), without any significant improvements seen in radiographic progression-free survival, time to castration-resistant disease, or overall survival [20]. The more recently published External Beam Radiation to Eliminate Nominal Metastatic Disease (EXTEND) trial is a phase 2, basket randomized clinical trial that aimed to evaluate the addition of metastasis-directed therapy (MDT) to hormone therapy for oligometastatic PCa. The study found that the combination of MDT with intermittent hormone therapy significantly improved progression-free survival (PFS) and eugonadal PFS compared to hormone therapy alone in men with oligometastatic PCa [21]. PSMA PET/CT has become the examination of choice in managing BCR and has changed the management approach in more than half of the cases with PSA recurrence [22,23].

Our results confirm that patients belonging to the BCR high-risk group are at a higher risk (59%) of having detectable lesions on PSMA PET/CT than patients belonging to the BCR low-risk group (36%). Among positive patients with distant lesions in the low-risk group, 100% were oligometastatic compared to 81% in the high-risk group. This stands in line with the available literature tackling this topic. Ferdinandus et al. demonstrated in their study that included 1960 patients the validity of the EAU risk classification by confirming that EAU BCR high-risk groups have higher rates of metastatic disease on PSMA PET/CT than low-risk groups. The striking finding was the significant percentage of patients with metastatic disease initially assigned to the low-risk group. In their study, 24% of patients included in the low-risk group had metastatic disease [24]. Indeed, Ferdinandus et al. argued that the PSMA PET/CT accurately represents the diverse extent of the disease, which is consistent with the actual clinical situation. This means that patients who have undergone radiotherapy or radical prostatectomy and belong to the same BCR risk group may experience different outcomes. Dong et al. demonstrated that even patients with low-risk BCR had relatively high detection rates on PSMA PET/CT with similar extra pelvic disease rates between high and low-risk groups [25]. Our study demonstrated comparably the important prevalence of oligometastatic disease in the low-risk group. Approximately a third of BCR low-risk patients with positive PSMA PET/CT were found to have distant metastatic disease. Such discordant grouping validates the inclusion of PSMA PET/CT within the risk assessment, the extent of which remains to be confirmed.

In an attempt to provide insight into the predictive and prognostic properties of PSMA PET/CT, Roberts et al. demonstrated that patients with regional or distant metastatic PSMA PET/CT findings were more likely to result in an event. An event was defined as a PSA > 0.2 ng/mL above the nadir following SRT or the use of additional hormonal and/or radiotherapy (RT). PSMA PET/CT scans were both prognostic and more accurate than the EAU risk classification in predicting the outcomes of sRT in patients who had undergone RP and were experiencing BCR. The EAU risk grouping and PSMA PET/CT results worked together to provide better predictions of outcomes for these patients [26]. Furthermore, Emmet et al. showed that the results of PSMA PET/CT scans are extremely effective in predicting the freedom from progression (FFP) at 3 years for men who have undergone sRT for BCR after RP. Men with negative PSMA PET/CT results or those whose disease is limited to the prostate area have a high FFP rate, despite receiving less extensive radiotherapy and lower rates of additional androgen deprivation therapy compared to those with a disease that has spread beyond the prostate [27]. Our results demonstrated a significantly higher percentage of local disease within the BCR low-risk group (26% vs. 6%, *p* < 0.001).

The predominant recurrence pattern across both risk groups in this study was regional recurrence (43% vs. 42%, *p* = 0.5). Rogowski et al. demonstrated the benefit of PSMA-PET/CT-based salvage elective nodal radiotherapy (sENRT) for LN recurrence, which translated into an improved biochemical recurrence-free survival and distant metastasis-free survival [28]. This sheds light on the clinical value of finding positive regional lesions on PSMA PET/CT and its implication in patients with discordant PSMA findings and EAU risk classifications.

BCR-free survival is significantly correlated with the initial PSA level, pathological Gleason score, presence of seminal vesicle invasion, extraprostatic extension, and intraductal pathology [29]. As part of the analysis by logistic regression, lymph node status (N0 vs. N1) at the time of radical prostatectomy was not found to be a statistically significant predictive factor for PSMA PET/CT positivity (OR1.4, *p* = 0.248). Nonetheless, the majority of N1 patients are more likely to have received adjuvant RT after radical prostatectomy, which might affect the disease progression.

PSA biochemical persistence (BCP) refers to a distinct form of relapse in which the levels of PSA remain elevated. This particular pattern of relapse has been linked to poorer cancer-related outcomes and is therefore not categorized into specific risk groups [24]. To avoid confounding factors and skewing results, we have excluded patients with persistent PSA post-radical prostatectomy.

In addition to the BCR risk group, PSA levels at the time of PSMA PET/CT and PSAvel were shown to be independent predictors of positivity. This agrees with recent findings demonstrating that higher PSA levels at the time of imaging and BCR high-risk group are predictive factors for PSMA PET M1 disease [24]. The PSA value at the time of PSMA PET/CT has already been validated as an independent predictor of PSMA PET/CT positivity [30]. The available data show a valid trend for including PSA velocity in the risk stratification algorithm for better patient risk evaluation of biochemical recurrence.

We acknowledge that limitations exist within this study, mostly emerging from its retrospective monocentric nature, the limited number of patients, and the lack of a comparison arm. Direct histological validation was rarely obtained. This is a known limitation in imaging studies, especially in recurrent PCa, as the biopsy of all PET-positive lesions is generally neither technically nor ethically feasible. Our analysis was performed on patients undergoing RP; therefore, these results could not be extrapolated to patients who underwent radiotherapy as the primary treatment modality. In the group of patients with PSMA-PET/CT performed before sRT, possible underestimation of the local prostate bed relapse cannot be excluded due to the physiologic urinary excretion of ^68^Ga-PSMA-11. All PSMA PET/CT scans were analyzed by a highly specialized, high-volume nuclear physician, limiting the generalizability of such findings.

## 5. Conclusions

This study confirms that the EAU BCR risk groups have different rates of PSMA PET/CT positivity. In the BCR low-risk group, one-third of patients had lesions on PSMA PET/CT. Out of those in the BCR low-risk group that presented with distant metastasis, all were oligometastatic. Such patients may benefit from metastasis-directed therapy, limiting the toxicity of systemic treatments. Given the presence of discordant positivity and risk classification, integrating PSMA PET/CT positivity predictors into risk calculators for BCR might improve patient classification for subsequent treatment options. Future prospective studies are still needed to validate the above findings and assumptions.

## Figures and Tables

**Figure 1 cancers-15-02926-f001:**
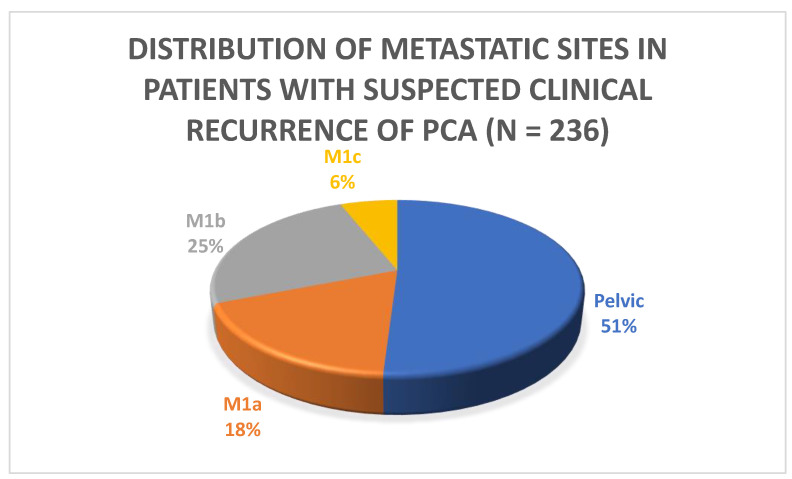
Distribution of metastatic sites in patients with suspected clinical recurrence.

**Figure 2 cancers-15-02926-f002:**
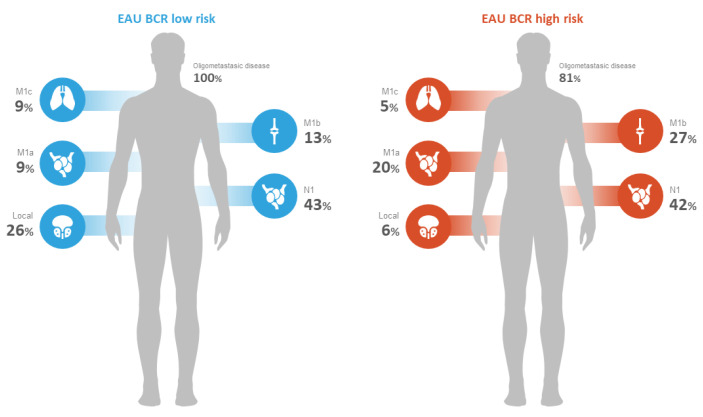
PSMA PET/CT disease extent according to EAU BCR risk groups.

**Figure 3 cancers-15-02926-f003:**
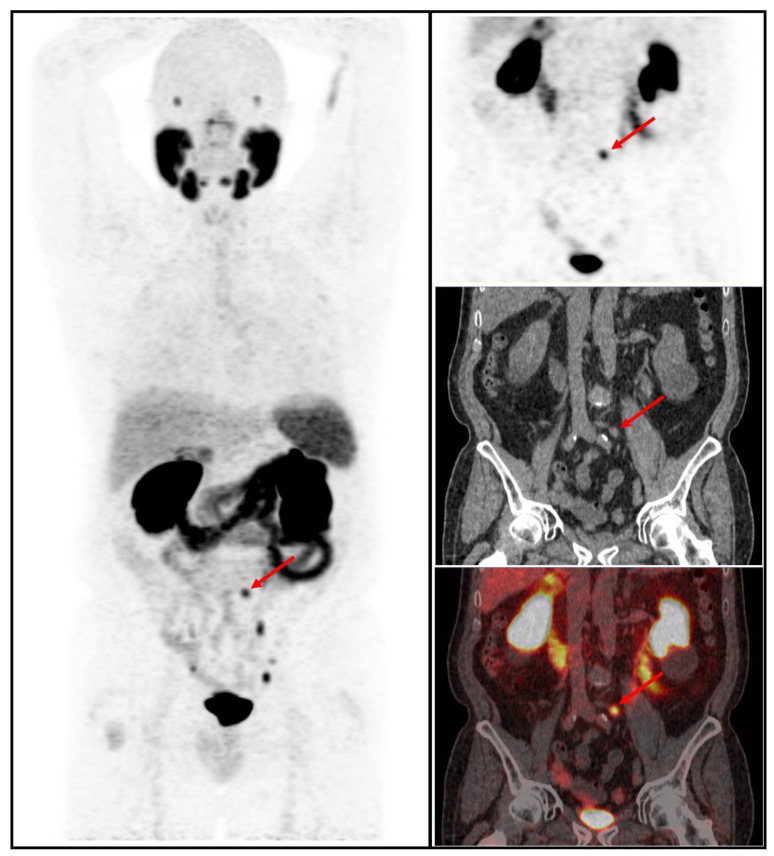
^68^Ga-PSMA-11 PET/CT whole body MIP, coronal PET, low-dose CT, and fused PET/CT images in a patient with BCR low risk showing increased focal uptake in the pelvic area corresponding to an infracentimetric left common iliac lymph node (red arrow).

**Table 1 cancers-15-02926-t001:** Definition of risk groups for biochemical recurrence by the EAU stratified by radical treatment.

Risk Group	Characteristics
BCR after RP	
Low risk	PSAdt > 12 months and ISUP < 4
High risk	PSAdt < 12 months and/or ISUP 4/5
BCR after radiation therapy	
Low risk	TTR > 18 months and ISUP < 4
High risk	TTR < 18 months and ISUP 4/5

BCR = Biochemical recurrence; PSAdt = PSA doubling time; ISUP = International Society of Urological Pathology; TTR = Time to recurrence.

**Table 2 cancers-15-02926-t002:** Population characteristics (*n* = 435).

Characteristics		Values
Age (years)		71 (65–75)
PSA at PET/CT (ng/mL), med (IQR)		1.14 (0.5–3)
PSAdt (months), med (IQR)		6.7 (3.9–13)
PSAvel (ng/mL/an), med (IQR)		0.9 (0.3–2.8)
pTstage, *n* (%)		
	T1c	2 (0.5)
	T2a	16 (3.7)
	T2b	39 (9)
	T2c	132 (30)
	T3a	155 (36)
	T3b	84 (19)
	Unknown	7 (1.6)
pN stage, *n* (%)		
	N1	40 (9.2)
	N0	222 (51)
	Nx	173 (40)
ISUP grade group, *n* (%)		
	1	69 (16)
	2	160 (37)
	3	117 (27)
	4	66 (15)
	5	23 (5.3)
Surgical margins, *n* (%)		
	R1	142 (33)
	R0	178 (41)
	Rx	115 (26)
Pelvic lymph node dissection, *n* (%)		
	Performed	243 (56)
	Not performed	126 (29)
	Unknown	66 (15)
Adjuvant RT, *n* (%)		122 (28)
Salvage radiotherapy, *n* (%)		200 (46)
BCR status, *n* (%)		
	First BCR	235 (54)
	Salvage RT	200 (46)
EAU BCR risk group, *n* (%)		
	Low risk	97 (22)
	High risk	338 (78)

PSAdt: PSA doubling time; PSAvel: PSA velocity; ISUP: International Society of Urological Pathology; RT: Radiotherapy; BCR: Biochemical recurrence.

**Table 3 cancers-15-02926-t003:** PSMA PET/CT results according to the EAU BCR risk groups.

		BCR Low Risk *n* = 97	BCR High Risk *n* = 338	Total *n* = 435	*p*-Value
Positivity rate *n* (%)		35 (36.08%)	201 (59%)	236 (54%)	<0.001
Number of distant metastatic lesions, *n* (%) *	1–5	11 (100%)	85 (81%)	96 (83%)	<0.001
	>5	0 (0%)	20 (19%)	20 (17%)	<0.001
Pattern of recurrence, *n* (%) **	Local N1	9 (26%) 15 (43%)	12 (6%) 84 (42%)	21 (8.9%) 99 (42%)	<0.001 0.5
	M1a	3 (9%)	40 (20%)	43 (18%)	0.008
	M1b	5 (13%)	54 (27%)	59 (25%)	0.007
	M1c	3 (9%)	11 (5%)	14 (6%)	0.96

* Percentage relative to patients with distant metastasis. ** Percentage relative to patients with a positive PSMA PET/CT.

**Table 4 cancers-15-02926-t004:** Analysis by logistic regression.

	Univariable Analysis		Multivariate Analysis	
	OR (95% CI)	*p*-Value	OR (95% CI)	*p*-Value
Positive predictive factors of PSMA PET/CT				
EAU BCR risk group (low vs. high)	2.5 (1.6–4)	<0.001	2.2 (1.3–3.8)	0.004
Tumor stage (≥T3a vs. <T3a)	1.7 (1.1–2.5)	0.009	1.4 (0.9–2.2)	0.109
Lymph node staging (N0 vs. N1)	1.4 (0.7–3)	0.248	-	-
Surgical Margins	1.3 (0.8–2.1)	0.207	-	-
Lymph node dissection (yes/no)	1.3 (0.8–2)	0.224	-	-
Adjuvant Radiotherapy (yes/no)	2 (1.3–3.2)	0.002	1.5 (0.9–2.5)	0.094
sRT (yes/no)	1.2 (0.8–1.8)	0.289	-	-
PSA level before PET/CT (≥0.5 vs. <0.5 ng/mL)	2.7 (1.7–4.2)	<0.001	2.7 (1.7–4.4)	<0.001
PSAvel (≥0.4 vs. <0.4 ng/mL/year)	2.6 (1.7–4)	<0.001 *	-	-
PSAdt (≥4 vs. <4 months)	0.5 (0.3–0.8)	0.002	0.8 (0.5–1.3)	0.354
Positive predictive factors of PSMA PET/CT in the BCR low-risk group				
Tumor stage (≥T3a vs. <T3a)	0.8 (0.4–2)	0.773		
Lymph node staging (N0 vs. N1)	2 (0.1–35.3)	0.616		
Surgical Margins	0.6 (0.2–1.6)	0.331		
Lymph node dissection (yes/no)	0.7 (0.3–1.7)	0.437		
Adjuvant Radiotherapy (yes/no)	2.7 (0.9–8)	0.079		
sRT (yes/no)	1.3 (0.6–3)	0.499		
PSA level before PET/CT (≥0.5 vs. <0.5 ng/mL)	3.9 (1.5–10.2)	0.006		
PSAvel (≥0.4 vs. <0.4 ng/mL/year)	5.8 (2–17)	0.001 *		
PSAdt (≥4 vs. <4 months)	1 (0.9–1)	0.092		

sRT: Salvage Radiotherapy; PSAvel: PSA velocity; PSAdt: PSA doubling time; OR: odds ratio; CI: Confidence interval. The method for determining the cut-off is detailed in Appendix A. * The PSAvel was excluded from the multivariable analysis to avoid collinearity effect (*p* = 0.949).

## Data Availability

Data are available on request due to restrictions (privacy or ethical). The data presented in this study are available on request from the corresponding author.

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
