# Peer review of "Peering through the PSMA PET Lens: The Role of the European Association of Urology Biochemical Recurrence Risk Groups after Radical Prostatectomy"

_cancers, 2023, doi:10.3390/cancers15112926_

Round 1

Reviewer 1 Report

Is an interesting paper about of the positivity rate of 68Ga-PSMA-11 PET/CT across BCR low and high risk groups and to establish positivity predictors. The paper is well written and the experiments are well conducted, however some points could be discussed. Some examples about of PSMA imaging could be provided. I suggest, include some discussion about of the lymph node positive and negative and the correlation between  BCR after RP, also include  the role of residual disease. Some representative figures are needed on the imaging findings part. I suggest improved the discussion about of the communication between the tumor and the lymph node to determine oligometastasis or distan metastasic disease. With all these data is posible to determine a ROC curve ? or if is the case Kaplan-Mayer graphics to compare the survival between low and high risk?. After revision the paper could be published.

Author Response

We would like to express our sincere gratitude for taking the time to review our manuscript. Your expertise and insightful feedback have been instrumental in shaping the final version of our paper

We herein provide a point by point response to your comments.

  1. Some examples about of PSMA imaging could be provided.

Response: We have taken this comment into consideration and have added a figure to our manuscript:

Figure 3: 68Ga-PSMA-11 PET/CT whole body MIP, coronal PET, low-dose CT and fused PET/CT images in a patient with BCR low risk showing an increased focal uptake in the pelvic area corresponding to an infracentimetric left common iliac lymph node (red arrow).

  1. Include some discussion about of the lymph node positive and negative and the correlation between BCR after RP

Response: We have carefully considered your suggestions to enhance the discussion section. We have included the following in the discussion section.

BCR free survival is significantly correlated with the initial PSA level, pathological gleason score, presence of seminal vesicle invasion, extraprostatic extension and intraductal pathology. As part of analysis by logistic regression, lymph node status (N0 vs N1) at the time of radical prostatectomy was not found to be a statistically significant predictive factor for PSMA PET/CT positivity (OR1.4, p=0.248). Nonetheless, the majority of N1 patients are more likely to have received adjuvant RT after radical prostatectomy which might affect the disease progression.

  1. Include the role of residual disease.

Response: Thank you for this comment. We have included a paragraph in the discussion section to clarify this point

PSA biochemical persistence (BCP) refers to a distinct form of relapse in which the levels of prostate-specific antigen (PSA) remain elevated. This particular pattern of relapse has been linked to poorer cancer-related outcomes and is therefore not categorized into specific risk groups. In order to avoid confounding factors and skewing of results we have excluded patients with persistent PSA post radical prostatectomy.

  1. Some representative figures are needed on the imaging findings part.

Response: We fully agree with the reviewer on this point. For better clarity a pie chart was included in the revised manuscript representing the distribution of PET positivity across all patients with positive PET PSMA. Figure 2 as indexed in the text demonstrates the distribution of positive lesions according to BCR risk group.

  1. With all these data is it possible to determine a ROC curve? Kaplan-Mayer graphics to compare the survival between low and high risk?

Response: This represents a very valid comment that we are seeking to study. Unfortunately, the data was not aimed at prognostic end points. This prognostic utility of a positive PET PSMA in low risk vs high risk BCR is an ongoing project. With the data available no prognostic end points were set for survival comparison. Event free survival seems to be a predictive prognostic marker to be used in future research.

Reviewer 2 Report

This is a well-written retrospective study evaluating the utility of PSMA-PET for staging patients after biochemical recurrence (BCR) post-radical prostatectomy (RP). The paper provides additional knowledge that nearly all patients with EAU low risk disease have oligometastatic disease. As stated by the authors, this is significant, due to the ability to incorporate metastasis-directed therapy into the treatment regimen. 

The authors do include patients with first and second BCR. In Table 2, there is mention of 235 patients with a first BCR, and 200 patients with salvage RT. Are the 122 patients with adjuvant RT included in the first BCR? 

From Table 4, it appears that adjuvant RT is associated with a higher risk cohort, based on the HR of 2 on univariate analysis. What were the general criteria for adjuvant treatment? 

Could the authors provide additional information on the distribution of the oligometastatic sites? (extra-pelvic nodal, bone, pulmonary)? This could be useful information for metastasis-directed therapy. 

Author Response

We would like to express our sincere gratitude for taking the time to review our manuscript. Your expertise and insightful feedback have been instrumental in shaping the final version of our paper

We herein provide a point by point response to your comments,

  1. The authors do include patients with first and second BCR. In Table 2, there is mention of 235 patients with a first BCR, and 200 patients with salvage RT. Are the 122 patients with adjuvant RT included in the first BCR? 

Response: This is a very good point.

The 122 patients with adjuvant RT includes all patients that underwent adjuvant RT during their disease course.

This includes patients n=100 that underwent adjuvant RT and experienced their first biochemical recurrence hence included in the 235 first BR group, and patients n=22 who underwent adjuvant RT followed by salvage Radiotherapy and included for a second biochemical recurrence.

  1. From Table 4, it appears that adjuvant RT is associated with a higher risk cohort, based on the HR of 2 on univariate analysis. What were the general criteria for adjuvant treatment? 

Indeed. The decision for adjuvant treatment is based on a multidisciplinary meeting in the presence of urologists, radiotherapists, medical oncologists, pathologists and radiologists.

The decision is usually based on patient and disease characteristics. However, the general criteria for adjuvant treatment includes

Aggressive pathology defined as ISUP>3

pT3 disease

Positive surgical margin

Patients with positive nodal disease

  1. Could the authors provide additional information on the distribution of the oligometastatic sites? (extra-pelvic nodal, bone, pulmonary)? This could be useful information for metastasis-directed therapy

This would be an interesting information for MDT as the reviewer has stated. Unfortunately for this study, the data we have so for lacks the distribution of the oligometastatic sites for the distant metastasis. However, extra pelvic nodal metastasis was reported as M1a.

A future study whose data is being matured, will evaluate the prognostic value of PET PSMA in such patients. The localization of oligometastatic sites will be reported in future publications.

We thank you for your insightful and well placed comments.